# 2-methylquinazoline derivative F7 as a potent and selective HDAC6 inhibitor protected against rhabdomyolysis-induced acute kidney injury

Jing Liu[1]◐, Xue Cui[2]◐, Fan Guo[1], Xinrui Li[1], Lingzhi Li[1], Jing Pan[1], Sibei Tao[1], Rongshuang Huang[1], Yanhuan Feng[1], Liang Ma◐[1]*, Ping Fu[1]

**1** Division of Nephrology and National Clinical Research Center for Geriatrics, Kidney Research Institute, West China Hospital of Sichuan University, Chengdu, China, **2** State Key Laboratory of Biotherapy and Cancer Center, West China Hospital, Sichuan University and Collaborative Innovation Center of Biotherapy, Chengdu, China

◐ These authors contributed equally to this work.

* liang_m@scu.edu.cn

**Data Availability Statement:** All relevant data are within the manuscript and its Supporting Information files.

## Abstract

Histone deacetylases 6 (HDAC6) has been reported to be involved in the pathogenesis of rhabdomyolysis-induced acute kidney injury (AKI). Selective inhibition of HDAC6 activity might be a potential treatment for AKI. In our lab, N-hydroxy-6-(4-(methyl(2-methylquinazo-lin-4-yl)amino)phenoxy)nicotinamide (F7) has been synthesized and inhibited HDAC6 activity with the $IC_{50}$ of 5.8 nM. However, whether F7 possessed favorable renoprotection against rhabdomyolysis-induced AKI and the involved mechanisms remained unclear. In the study, glycerol-injected mice developed severe AKI symptoms as indicated by acute renal dysfunction and pathological changes, accompanied by the overexpression of HDAC6 in tubular epithelial cells. Pretreatment with F7 at a dose of 40 mg/kg/d for 3 days significantly attenuated serum creatinine, serum urea, renal tubular damage and suppressed renal inflammatory responses. Mechanistically, F7 enhanced the acetylation of histone H3 and α-tubulin to reduce HDAC6 activity. Glycerol-induced AKI triggered multiple signal mediators of NF-κB pathway as well as the elevation of ERK1/2 protein and p38 phosphorylation. Glycerol also induced the high expression of proinflammatory cytokine IL-1β and IL-6 in kidney and human renal proximal tubule HK-2 cells. Treatment of F7 notably improved above-mentioned inflammatory responses in the injured kidney tissue and HK-2 cell. Overall, these data highlighted that 2-methylquinazoline derivative F7 inhibited renal HDAC6 activity and inflammatory responses to protect against rhabdomyolysis-induced AKI.

## Introduction

Acute kidney injury (AKI) consists of a series of heterogeneous conditions characterized by a sudden decline in glomerular filtration and oliguria [1], associated with high risk of chronic

**Funding:** LM was funded by Key R&D Program of Sichuan Province (2018FZ0104), Program of National Clinical Research Center for Geriatrics, West China Hospital of Sichuan University (Z2018B15) and Innovation Program of Sichuan University (2018SCUH0077). The funders had no role in study design, data collection and analysis, decision to publish, or preparation of the manuscript.

**Competing interests:** The authors have declared that no competing interests exist.

kidney disease (CKD) and high mortality [2]. Rhabdomyolysis, mainly induced by trauma, exertion, muscle hypoxia, infections, drugs and toxins, etc, is one of important causes leading to AKI[3]. Rhabdomyolysis-induced AKI is common but life-threatening, accounting for 5–15% of all community-acquired AKI cases [4, 5]. However, the detailed mechanism of rhabdomyolysis-AKI is poorly elucidated and treatment strategies remain limited.

Considerable evidences suggested that direct and ischemic tubular injury played a key role in myoglobin-induced AKI [6], and cellular damage is considered as main triggers initiating inflammatory response in acute tissue injury[7]. Multiple inflammatory genes and signaling pathways were activated in rhabdomyolysis-induced AKI [8]. A number of studies demonstrated that nuclear factor-kappa B (NF-κB) played crucial roles in modulating inflammation and cell proliferation in renal diseases [9–12]. NF-κB normally existed in cytoplasm in inactive form binding to its inhibitor, immuno-globulin (Ig) κ light chain gene of B lymphocytes (IκB) [13]. Stimuli could elicit the activation of IκB, thus inducing NF-κB released from IκB, and translocated into nucleus to promote transcriptional activity.

HDACs are a group of enzymes acting as removing acetyl group from lysine residues in histone or nonhistone proteins. HDACs are categorized into 4 classes based on homology: class I (HDAC 1, 2, 3, and 8), class II (HDAC 4, 5, 6, 7, 9 and 10), class III (SIRT 1–7) and class IV (HDAC 11)[14]. HDAC6 stands out because of its almost exclusive deacetylation in cytoplasm and the involvement of both dependent and independent of its catalytic activity, which opened the way for the identification of its substrates and development of highly selective inhibitor of its enzyme action [15]. Meanwhile, the elevated expression of histone deacetylase 6 (HDAC6) have been confirmed to be involved in renal diseases including AKI [16–18], autosomal dominant polycystic kidney disease [19], and hypertensive nephropathy [20] for the contribution to inflammation, apoptosis and fibrosis.

We previously reported that a selective HDAC6 inhibitor 23BB possessed renoprotective effects by improving tubular cell apoptosis and down-regulating endoplasmic reticulum stress [16]. Recently, N-hydroxy-6-(4-(methyl(2-methylquinazolin-4-yl)amino)phenoxy) nicotinamide (F7) has been designed, synthesized in our lab and inhibited HDAC6 activity with the $IC_{50}$ of 5.8 nM. However, whether novel HDAC6 inhibitor F7 possessed favorable renoprotection against rhabdomyolysis-induced AKI and the involved mechanisms remained unclear.

## Material and methods

### Ethical statement

All applicable international, national, and/or institutional guidelines for the care and use of animals were followed. All procedures performed in studies involving animals were in accordance with the ethical standards of animal ethics committee of Sichuan University (IACUC number: 2017080A).

### Animals

Male C57BL/6 mice (8 weeks; 25-28g) were purchased from the Animal Laboratory Center of Sichuan University (Chengdu, China). These mice were housed in the controlled environment with constant temperature at 23 ± 2˚C, humidity at 50–60%, and 12-hour light/dark cycle, and had free access to water and food. All the experiment procedures were in accordance with The Guide for Care and Use of Laboratory Animals, and the study protocol was approved by the Animal Care and Use Committee of Sichuan University (IACUC number: 2017080A).

## Rhabdomyolysis-induced AKI model and pretreatment

The synthesis of F7 was in six synthetic steps with moderate yield according to the previous procedures of selective HDAC6 inhibitor 23bb[21]. HDACs inhibition assay of F7 was performed utilizing 4-amino-7-methylcoumarin (AMC) labeled Ac-peptide (Ac-peptide-AMC) substrates, a service provided by Chempartner Company (Shanghai, China). Results of $IC_{50}$ and selectivity of F7 against HDAC1-11 enzymes were all listed in S1 Fig.

After one week of adoption to the housing conditions, the mice were randomly divided into three groups (n = 8): control, glycerol, and glycerol+HDAC6 inhibitor (F7). For the latter two groups, a single intramuscular injection of 50% glycerol dissolved in 0.9% normal saline (10 μl/g) was divided between bilateral back limbs to induce rhabdomyolysis-induced AKI model. The mice in control group received the same dose of saline injection at the place of glycerol. Before glycerol injection, mice in pretreatment group were orally administrated with F7 (dissolved in DEG and diluted with 0.9% saline) at a dose of 40mg/kg/d (200 μl solution for 28g) for 3 days. For control and glycerol groups, mice were treated with the same dose of saline.

All the mice were euthanized via pentobarbital sodium injection (50 mg/kg, i.p.). Terminal blood samples were collected and serum samples were stored at -80˚C. Kidney tissues were divided and respectively removed into 10% phosphate buffered formalin for histological staining, and into liquid nitrogen follows by storage at -80˚C for further quantitative analysis and immunofluorescence staining.

## Biochemical evaluation of blood sample

The serum level of creatinine kinase (CK), blood urea (UREA) and serum creatinine (sCr) were tested using high performance liquid chromatography (HPLC) by Institute of Drug Clinical Trial and the GCP center of West China Hospital of Sichuan University.

## Histological examination

Paraffin-embedded kidney tissue samples were cut into 4-um sections and stained with haematoxylin and erosin (H&E) and periodic acid-Schiff (PAS) staining. These sections were viewed using light microscopy at magnifications of ⊆200 and ⊆400 respectively, and semi-quantitative estimation of renal tubular damage was performed on a scale from 0 to 4 to grade the percentage of injured/damaged renal tubules: 0 for 0%, 0.5% for <10%, 1 for 10–25%, 2 for 26–50%, 3 for 51–75% and 4 for 76–100%[22]. At least 10 areas were randomly selected in the cortex per slide.

## Immunofluorescence staining

Kidney tissue samples were mounted in O.C.T. compound medium (Tissue-Tek) for cryosectioning (4 μm). Sections were immediately fixed in 10% buffered formalin, washed, dehydrated and stored at -20˚C. Before immunofluorescence staining, the tissue sections were rehydrated and blocked with 10% horse medium (diluted with PBS solution) for 1 hour at room temperature and incubated at 4˚C overnight with indicated primary antibodies. Then sections were exposed to second antibodies labeled with Cy5 red or FITC green (Jackson ImmunoResearch Inc., West Grove, PA, United States). The nuclei were counterstained with DAPI (1:500, Life Technologies Corporation, OR, United States). The images were captured with AxioCam HRc digital camera (Carl Zeiss).

## Western blotting

Kidney tissue or cultured cell samples were sufficiently lysed in the mixture of RIPA buffer (50 mM Tris (pH7.4), 150 mM NaCl, 1% TritonX-100, 1% sodium deoxycholate, 0.1% SDS) containing 4% cocktail proteinase inhibitor (Roche, Switzerland) on ice. After centrifuged at 13,000 rpm for 15 min at 4°C, supernate of the homogenate were collected for protein quantification. Protein equivalent to 50 ug were loaded into 10–12% SDS-PAGE gels for electrophoretic separation, and then transferred to polyvinylidene difluoride (PVDF) membranes (0.2 μm, Bio-Rad Laboratories, Inc.). After protein blotting, membranes were blocks using 5% non-fat dried milk for 1h at room temperature, then probed with primary antibodies at 4°C overnight, and incubated with secondary antibodies for 1h at room temperature. Primary antibodies included: mouse anti-GAPDH (1:2000, 200306-7E4, Zen BioScience Inc., Research Triangle Park, NC, USA), mouse anti-IL-6 (1:1000, EM170414, HuaAn Biotechnology Co., Ltd, Hangzhou, China), mouse anti-HDAC6 (1:200, sc-28386, Santa Cruz Biotechnology Inc., Dallas, TX, United States)), rabbit anti-acetyl-$H_3$ (1:1000, #9649), rabbit anti-IκBα (1:1000, #4812), rabbit anti-phospho-IκBα (1:1000, #2859), rabbit anti-NF-κB p65 (1:1000, #8242s), rabbit anti-phospho-NF-κB p65 (1:1000, #3031), rabbit anti-ERK1/2 (1:1000, #9102), and rabbit anti-phospho-ERK1/2 (1:1000, #8544) (all the seven were purchased from Cell Signaling Technology, Danvers, MA, USA), rabbit anti-alpha-tubulin (1:500, ab179484), rabbit anti-IL-1β (1:1000, ab9722) (both were obtained from Abcam, Cambridge, MA, USA) and rabbit anti-acetyl-alpha-tubulin (1:1000, DF2982, Affinity BioScience, Cincinnati, OH, United States). Secondary antibodies included goat anti-mouse and goat anti-rabbit IgG (1:2000, Biosynthesis Biotechnology Co., Ltd., Beijing, China). Immunoblots were visualized with the Immobilon Western Chemiluminescent HRP Substrate (Millipore Corporation, Billerica, MA, United States) and measured with Bio-Rad Chemi Doc MP.

## Quantitative real-time PCR analysis

Total RNA was obtained from frozen kidney tissue using the total RNA extraction Kit (BioTek, Winooski, VT, United States), and reverse transcription was performed using PrimeScript RT Reagent Kit (Takara Bio, Inc., Otsu, Japan) according to protocols. Reactions of PCR amplification were quantified using the iTaq Universal SYBR Green Supermix (Bio-Rad Laboratories, Inc.) in a PCR system (CFX Connect; Bio-Rad, Hercules, CA, United States). All results were presented with relative expression levels normalized to GAPDH. Primer sequences: IL-6 (Mouse): `Forward: 5'-ACAACCACGGCCTTCCCTACTT-3', Reverse: 5'-CACGA TTTCCCAGAGAACATGTG-3'; TNF-α (Mouse): Forward: 5'-ACCCTCACACTCA GATCATCTTC-3', Reverse: 5'-TGGTGGTTTGCTACGACGT-3'; NGAL(Mouse): Forward: 5'-GCAGGTGGTACGTTGTGGG-3', Reverse: 5'-CTCTTGTAGCTCATA GATGGTGC-3'.`

## HK-2 Cells, myoglobin administration and F7 treatment

Human renal proximal tubule cell line (HK-2) cells were purchased from Shanghai Institute of Biochemistry and Cell Biology, Shanghai, China and maintained in DMEM/F12 (Hyclone, Beijing, China) supplemented with 10% fetal bovine serum (FBS, Hyclone, Australia) and 1% penicillin and streptomycin at 37°C under humidified atmosphere of 5% carbon dioxide. Then cells were seeded on six-well plates (Shanghai Sunub Bio-Tech Development Inc., Shanghai, China) at a density of 200,000 cells per well, divided into four groups at exponential phase: the control group (cells incubated with DMEM/F12+10%FBS), the model group (cells incubated with 200 μM ferrous myoglobin (M0630, Sigma-Aldrich, St. Louis, MO, USA)), the

F7-1.25 nM group (cells incubated with ferrous myoglobin+F7 at 1.25nM) and the F7-10 nM group (cells incubated with ferrous myoglobin+F7 at 10 nM).

## Annexin V and PI staining

Cell apoptosis was evaluated by flow cytometry using an Annexin V-FITC apoptosis analysis kit (AO-2001-02P-H, Tianjin Sungene Biotech Co., Ltd, Tianjin, China). After the four group of cells were cultured under the condition as described above for 24 hours, they were digested by trypsin, collected, and washed three times with PBS diluted 1×binding buffer (1ml per tube). For each sample with 100 μL cell suspension, 5 μL (2.5 μg/mL) of Annexin-V-FITC were added, and after gentle vortex the samples were incubated for 10 min at room temperature in the dark. Then 5 μL of PI solution were added and incubated at the same condition. After being supplemented to 500 μL using cold PBS followed by gentle vortex, samples were ready to be testes. Apoptosis signals were detected using a CytoFLEX by Beckman & Coulter.

## Statistical analysis

Description data were presented as means ± standard deviation. A Student two-tailed unpaired t test was used for pairwise comparison. One-way ANOVA and further Turkey test were used for multiple comparisons. A two-sided p value less than 0.05 was considered statistically significant.

## Results

### Selective HDAC6 inhibitor F7 improved acute renal function and alleviated kidney damage in rhabdomyolysis-induced AKI

We firstly examined whether the mice model of glycerol-induced AKI was successfully established, and tested the protective effect of F7 at a dose of 40 mg/kg/d for 3 days on renal function and pathological changes. Serum creatine kinase (CK) in glycerol group and glycerol+F7 group was markedly elevated compared to that of control at 24 hours after glycerol injection (Fig 1A). The values of serum urea (UREA) and serum creatinine (sCr) in glycerol group were approximately five and three-fold more than that of control (Fig 1B and 1C), while the value of glycerol+F7 group was remarkably decreased compared to that of glycerol group. Consistent findings were obtained from histological changes in H&E and PAS-staining sections: serious tubular damages (tubular dilatation, swelling, necrosis and cast formation) were observed in glycerol group while pathological injuries were effectively alleviated by the pretreatment of F7 (Fig 1D and 1E). We also evaluated the liver, kidney, spleen, heart and lung tissues of mice with oral administration of F7 at the same dose as that in F7+glycerol group for 3 days, but without glycerol injections. Histological sections confirmed the F7 did not exhibited organ damage (S2 Fig).

### F7 inhibited HDAC6 activity in the kidney of rhabdomyolysis-induced AKI

Results of western blot and immunofluorescence demonstrated that HDAC6 minimally expressed in normal kidneys of control group, while HDAC6 level in the glycerol group was markedly increased. Pretreatment of F7 significantly inhibited the expression of HDAC6 in the damaged kidney. Since HDAC6 deacetylated both histone (i.e. histone H3[23]) and non-histone (i.e. α-tubulin[15, 24]), thus the inhibition of HDAC6 could be reflected by enhancing the acetylation of HDAC6 substrates. Protein expression and immunoblot analysis results were shown in Fig 2A. The high overlapping level of lectin and HDAC6 from merged

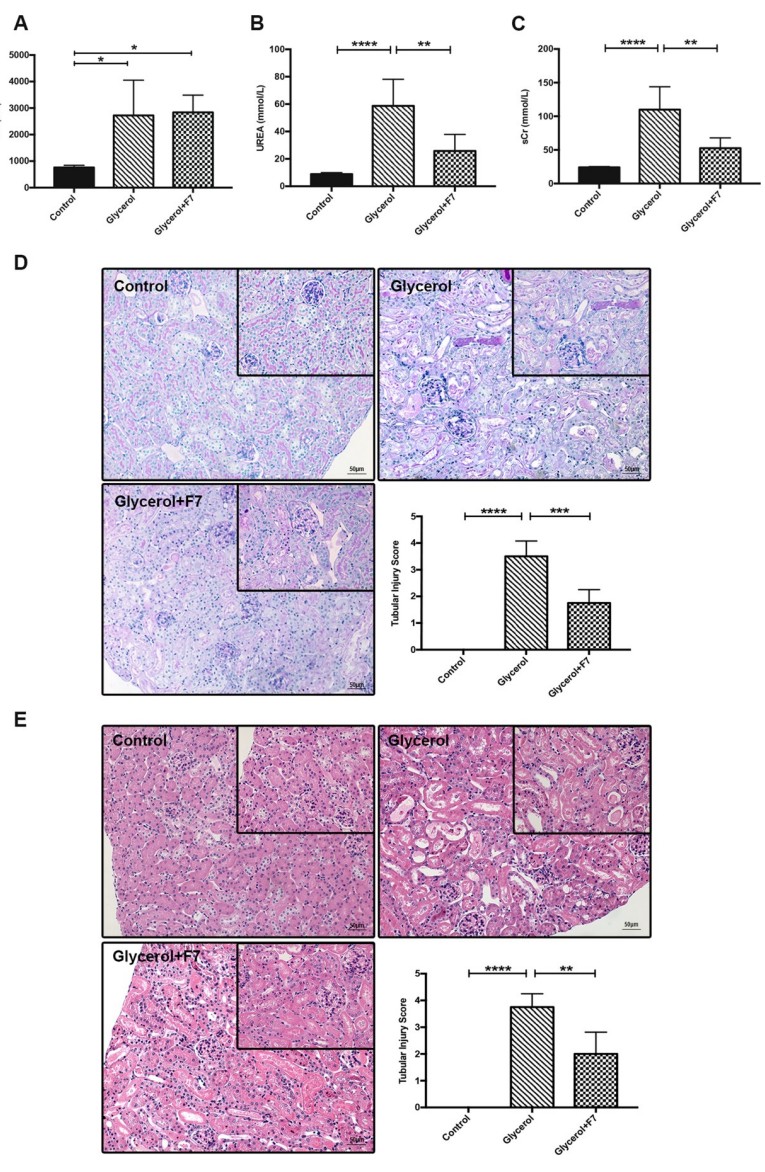

**Fig 1. F7 alleviated kidney injury in rhabdomyolysis-induced AKI.** (A) Serum creatine kinase (CK); (B) serum urea (UREA); (C)serum creatinine (sCr). (D) Photomicrographs (×200 and ×400) illustrated PAS staining of the kidney tissues, and the histogram presented a semiquantitative assessment of tubular injury. (E) Photomicrographs (×200 and ×400) illustrated H&E staining of the kidney tissues, and the histogram presented a semiquantitative assessment of tubular injury. N = 6, ****P<0.0001, ***P<0.001, **P<0.01, *P<0.05. Images (Fig 1D and 1E) are representative of more than 10 area views from each group (N = 6).

immunofluorescence results also indicated HDAC6 mainly overexpressed in tubular cells (**Fig 2B**), which was consistent with the finding that HDAC6 actively maintained in the cytoplasm [25].

## F7 attenuated tubular damage of rhabdomyolysis-induced AKI

Neutrophil gelatinase-associated lipocalin (NGAL) is a main biomarker of kidney tubular injuries, and the increase in NGAL production and release from tubular cells could be found after kidney receiving harmful stimuli [26]. We firstly found that the NGAL expression significantly

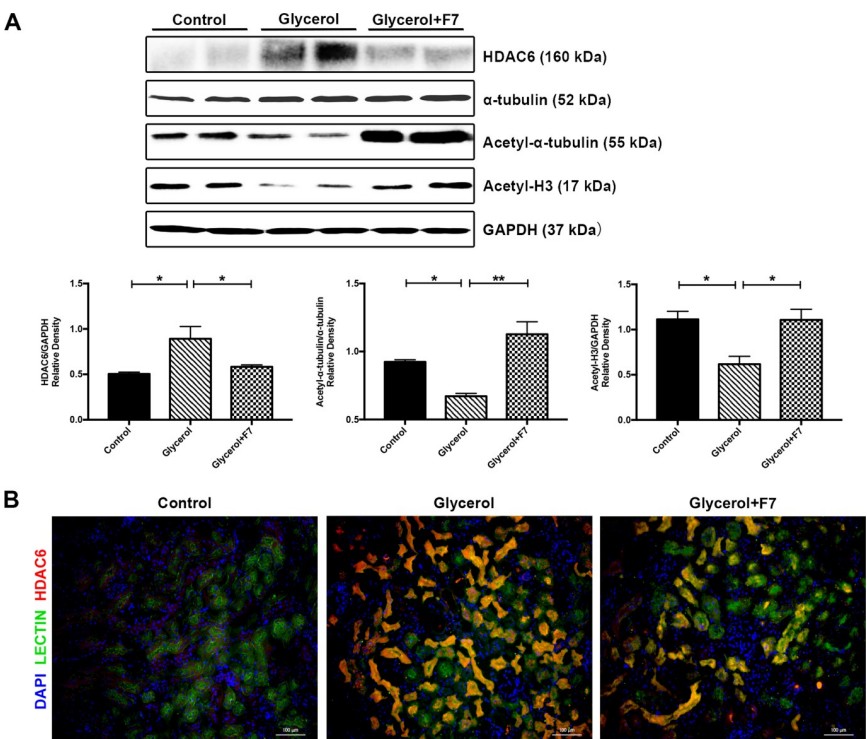

**Fig 2. F7 inhibited HDAC6 expression and enhanced the deacetylation of α-tubulin and histone H3 in the kidney of rhabdomyolysis-induced AKI.** (A) Protein expression level of HDAC6, α-tubulin, acetyl-α-tubulin and acetyl-H3. Immunoblot analysis of HDAC6 and its substrates. The densitometry values of acetyl-α-tubulin was normalized to α-tubulin. The densitometry values of HDAC6 and acetyl-H3 were normalized to GAPDH. N = 5, **P<0.01, *P<0.05. The data are representative of 2–3 independent experiments. (B) F7 inhibited expression of HDAC6 as measured by Double immunofluorescence staining of HDAC6 and lectin using kidney tissue frozen sections. Lectin was the marker of tubular epithelial cells. Images are representative of more than 10 area views from each group (N = 3).

increased and was mainly located in tubules of glycerol group by immunofluorescence staining, while the expression level was largely diminished by F7 treatment (**Fig 3A**). Renal protection by HDAC6 inhibition was further tested by real-time PCR and immunoblot analysis. NGAL mRNA expression and protein expression were both obviously elevated in glycerol group while pre-treatment of F7 markedly reduced expressions (**Fig 3B**).

## F7 pretreatment inhibited the activation of NF-κB and MAPKs, and ameliorated the levels of inflammatory cytokines in rhabdomyolysis-induced AKI

Inflammatory response played an essential role in the pathogenesis of AKI[27], and the activation of NF-κB is the critical mechanism mediating the expression of key inflammatory molecules in the process of rhabdomyolysis[28]. Data from **Fig 3C** demonstrated that the phosphorylation of IκBα was the highest in glycerol group, and pretreatment with F7 decreased the IκBα phosphorylation. Consistently, the increased phosphorylated NF-κB (p65) was found in glycerol group, and F7 effectively down-regulated the p65 phosphorylation. Moreover, total extracellular signal-regulated kinases (ERK1/2) expression and ERK1/2 phosphorylation level was upregulated in the glycerol group, together with obvious p38 phosphorylation. F7 effectively inhibited total ERK1/2 expression and p38 phosphorylation (**Fig 3D**).

From analysis of western blot, a dramatic increase of proinflammatory cytokines (IL-6 and IL-β) could be observed in glycerol group, and F7 remarkably down-regulated the levels, as

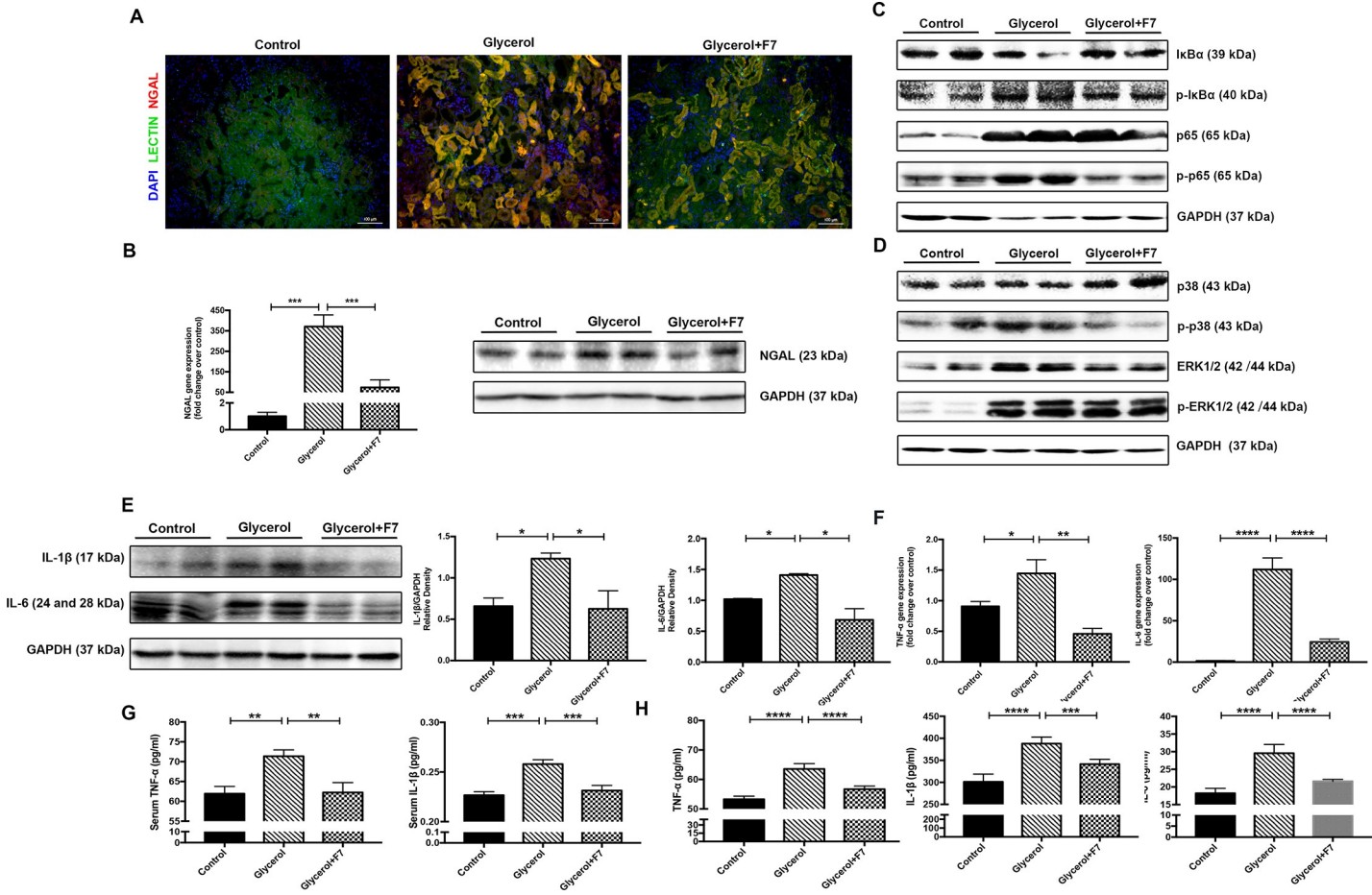

**Fig 3. F7 alleviated tubular damages, suppressed NF-κB and MAPK signaling pathway, and effectively attenuated inflammatory responses in rhabdomyolysis-induced AKI.** (A) Double immunofluorescence staining of gelatinase-associated lipocalin (NGAL) and lectin in kidney tissues (×200). Lectin was the marker of tubular epithelial cells. (B) Effect of F7 on tubular damages quantified by real-time PCR. (C) Expression level of key proteins of NF-κB signaling pathway in control, glycerol and glycerol+F7 group. (D) Protein expression level of p38 and ERK1/2 in control, glycerol and glycerol+F7 group. (E) Protein expression level of IL-1β and IL-6; Immunoblot of IL-1β and IL-6 normalized to GAPDH; (F) Real-time PCR analysis of TNF-α and IL-6. (G) ELISA analysis of serum proinflammatory cytokines (TNF-α and IL-β). (H) ELISA analysis of proinflammatory cytokines in kidney lysates (TNF-α, IL-1β and IL-6). ****P<0.0001, ***P<0.001, **P<0.01, *P<0.05. Images are representative of more than 10 area views from each group (N = 3). Data are representative of 2–3 independent experiments.

shown in **Fig 3E**. Similarly, the increased mRNA expression of IL-6 and TNF-α could be found in glycerol group compared to that of control, while F7 significantly decreased the mRNA level of these two cytokines (**Fig 3F**). Proinflammatory cytokines in kidney lysates and serum were also examined using ELISA. It could be found in both serum and kidney lysates that IL-β and TNF-α were obviously elevated in glycerol group while decreased in F7 treatment group. Additionally, similar change was observed in IL-6 from kidney lysates (**Fig 3G and 3H**).

## F7 improved cell growth and inflammatory cytokines by inhibiting NF-κB activity in HK-2 cells

To further verify the *in vitro* renoprotection of F7, we examined the pharmacological toxicity and effectiveness of F7 at a respective concentration of 2.5 nM, 5 nM and 10 nM, to select the optimal F7-treated concentration. Apoptosis signals from flow cytometry revealed that F7 improved growth of HK-2 cells under stimulation of myoglobin (Mb) in a dose-dependent

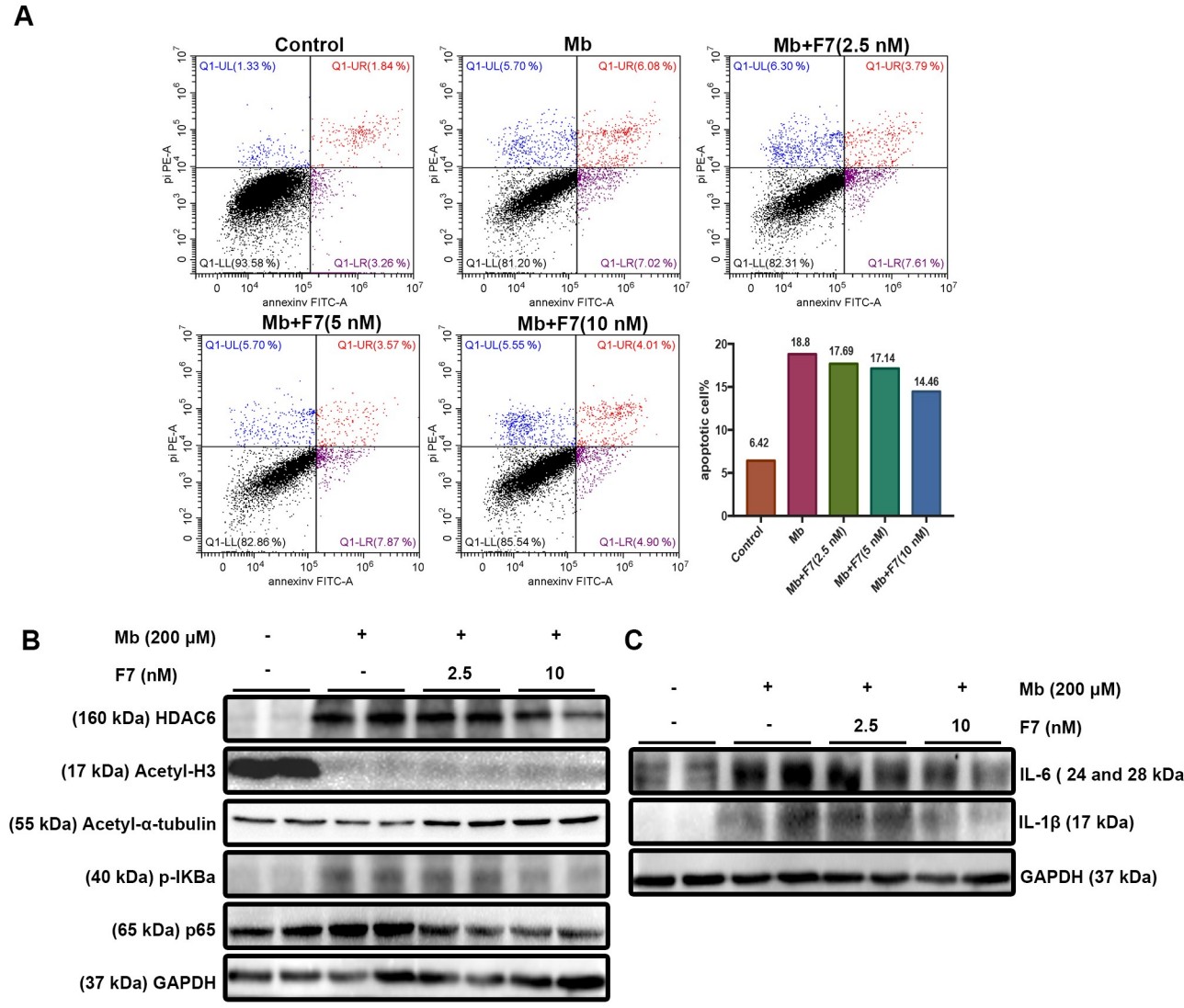

**Fig 4. F7 improved the cell growth and ameliorated myoglobin-induced inflammation in HK-2 cells.** (A) HK-2 cells in myoglobin group and glycerol+F7 group were stimulated by ferrous myoglobin (200 μM) for 24 hrs. Apoptosis due to myoglobin and treatment effect of F7 on cell growth were detected using flow cytometry, thus to select optimal F7-treated concentration. (B) F7 effectively inhibited HDAC6 expression in vitro; (C) F7 ameliorated inflammation in HK-2 cells by inhibiting HDAC6 and suppressing the activation of NF-κB pathway in vitro. Data are representative of 2–3 independent experiments.

manner (Fig 4A), thus indicated that the optimal F7-treated concentration was 10 nM. Results of western blot analysis indicated that successful inhibition of HDAC6 in vitro suppressed the phosphorylation of NF-κB (p65) and reduced the degree of inflammatory response compared to those of Mb group as shown in Fig 4B and 4C.

## Discussion

In our previous studies, we have confirmed that HDAC6 contributed to the pathogenesis of rhabdomyolysis-induced AKI. Selective inhibition of HDAC6 activity by a small-molecule compound N-hydroxy- 4-(2-methoxy-5-(methyl(2-methylquinazolin-4-yl)-amino)phenoxy) butanamide (23bb) might be a promising strategy for the treatment of AKI. Recently, N-hydroxy-6-(4-(methyl(2-methylquinazolin-4-yl)amino)phenoxy)nicotinamide (F7) has been

designed, synthesized in our lab and inhibited HDAC6 activity with the IC$_{50}$ of 5.8 nM. HDAC enzyme inhibition assay indicated that HDAC6 inhibition efficacy of F7 was three times higher than that of 23BB, and inhibition selectivity ratio of F7 (inhibition activity of HDAC6 vs other HDACs) were all much higher than that of 23BB (details were listed in **S1 Table**). Base on the good performance on inhibition efficacy and selectivity, we further investigated the renoprotective effect of F7 and the involved mechanisms in rhabdomyolysis-induced AKI. The overexpression of HDAC6 was observed in the injured kidneys of rhabdomyolysis-induced AKI. Pretreatment of F7 effectively improved renal functions, alleviated kidney histopathological damages, decreased the expression of kidney injury biomarkers, and suppressed NF-κB signaling pathway. The consistent findings also could be observed *in vitro* in HK-2 cells. These results strongly indicated that HDAC6 contributed to the development of inflammation in AKI induced by rhabdomyolysis through activating NF-κB signaling pathways.

In rhabdomyolysis-induced AKI, myoglobin plays the key role in leading to renal toxicity through multiple deleterious effects including tubular obstruction by myoglobin-derived casts, oxidative stress, inflammation, apoptosis and vasoconstriction [4]. Inflammation in AKI is known to be a complex biological process that is crucial to repairing injured tissue. As a key modulator, HDAC6 is involved in multiple biological processes ranging from gene expression to protein activity, thus participating in the inflammation. Recently, several studies demonstrated the increased expression of HDAC6 in cytoplasm of tubular epithelial cells in AKI induced by cisplatin [29] and rhabdomyolysis [16, 17], among which the elevated production of pro-inflammatory cytokines [17, 29] and apoptosis-related biomarkers [16] were observed. Under the treatment of HDAC6 inhibitor, inflammatory response and apoptosis were ameliorated accompanied by the decreased HDAC6 activity. The mechanism of HDAC6 regulating apoptosis through endoplasmic reticulum stress was evidenced in our previous study [16].

As a key transcription regulator of inflammation, NF-κB promotes the expression of pro-inflammatory cytokines and adhesion molecules. Both clinical and experimental data confirmed the elevation and activation of NF-κB in a variety of renal inflammatory disorders [12, 30, 31]. Meanwhile, it was found that NGAL could be induced to overexpress by NF-κB as a member of lipocalin superfamily and biomarker of AKI [32]. The close linking between HDAC6 and NF-κB was further strengthened by our study with the sharp elevation of NGAL in glycerol group and dramatic reduction in glycerol+F7 group. As for the mechanism of NF-κB activation, if receiving potent stimuli, IκBα could be rapidly degraded within minutes by phosphorylation, polyubiquitinylation and/or proteasome [33, 34]. Consequently, NF-κB was free and translocated into the nucleus where it bound and modulated the expression of target genes [35]. The signal is ultimately terminated by the new synthesis of IκB [12]. Our study demonstrated the increased phosphorylation of IκB and the elevated expression of p65 in the injured kidneys of rhabdomyolysis-induced AKI and in myoglobin-induced HK-2 cells. Additionally, total IκB expression was the least in glycerol group. After treatment of F7 both *in vivo* and *in vitro*, IκBα phosphorylation and activation of p65 were both obviously inhibited, accompanied by the markedly suppressed inflammation. All the findings highlighted that HDAC6 inhibitor regulated NF-κB pathway which participated in the rhabdomyolysis-induced AKI.

In the study, our data also revealed that F7 could alleviate inflammation through inhibiting the phosphorylation of IκBα and p65, but how HDAC6 interacted with IκBα, p65 and other NF-κB family factors was unclear. Previous study [11] found HDAC6 coimmunoprecipitated with NF-κB p50 and p65, and reported that p50 and p65 seemed to recruit HDAC6 from cytoplasm to nuclear DNA-protein complex. Although HDAC6 predominantly existed in cytoplasm, our results found that F7 induced the acetylation of α-tubulin and histone H3, which indicated that HDAC6 might be involved in multiple mechanisms and compartments to

mediate activation of NF-κB pathway. Moreover, NF-κB was unlikely to be the sole pathway interacting with HDAC6 to induce inflammatory response. MAPK and PI3K/Akt signaling pathways were also upregulated in rhabdomyolysis-induced AKI [8]. In our study, the overexpression of ERK1/2 and the activation of p38 phosphorylation could be found in glycerol group where HDAC6 protein was elevated, while no significant change was observed about c-Jun NH2-terminal kinase (JNK). These findings could be supported by the previous reports that ERK1/2 bound and phosphorylated HDAC6 [36], and HDAC6 activated the phosphorylation of p38 by deacetylation [37]. It was reported that the overexpression of HDAC6 could activate JNK and increase the phosphorylation of JNK in liver cancer cells [38]. Specifically, JNK1 was associated the with sustaining HDAC6 level in several organ-derived cell lines [39]. Of notes, the association between JNK and HDAC6 were not previously reported in kidney which needed to be further investigated.

In summary, the most significant thing was that we confirmed novel HDAC6 inhibitor F7 alleviated acute renal function against rhabdomyolysis-induced AKI and selective inhibition of HDAC6 activity was a promising strategy for the treatment of AKI. These findings also indicated that F7 suppressed HDAC6 activity to reduce inflammatory responses via the inhibition of NF-κB pathway in rhabdomyolysis-induced AKI.

## Supporting information

**S1 Fig. The IC$_{50}$ and selectivity of F7 against HDAC1-11 enzymes.**
(TIFF)

**S2 Fig. Histological examinations of different organs from the normal mice administrated with F7.** Photomicrographs (×200) illustrated H&E staining of the heart, liver, spleen, lung and kidney tissues from mice with oral administration of F7 at a dose of 40 mg/kg/d for 3 days but without glycerol injection.
(TIF)

**S1 Table. Comparison of HDAC Inhibition Activity among HDAC6 inhibitors.**
(PDF)

## Acknowledgments

The study was supported by Innovation Program of Sichuan University (2018SCUH0077), and Program of National Clinical Research Center for Geriatrics from West China Hospital of Sichuan University (Z2018B15).

## Author Contributions

**Conceptualization:** Jing Liu, Xue Cui, Liang Ma, Ping Fu.

**Data curation:** Jing Liu, Xue Cui, Xinrui Li.

**Formal analysis:** Jing Liu, Xue Cui.

**Funding acquisition:** Liang Ma.

**Investigation:** Jing Liu, Xue Cui, Fan Guo, Xinrui Li, Lingzhi Li, Jing Pan, Sibei Tao, Rongshuang Huang, Yanhuan Feng.

**Methodology:** Jing Liu, Fan Guo, Lingzhi Li, Jing Pan, Sibei Tao, Rongshuang Huang.

**Project administration:** Jing Liu, Xue Cui, Fan Guo.

**Software:** Jing Liu, Xinrui Li.

**Supervision:** Fan Guo, Yanhuan Feng, Liang Ma, Ping Fu.

**Validation:** Yanhuan Feng, Liang Ma, Ping Fu.

**Visualization:** Jing Liu, Lingzhi Li.

**Writing – original draft:** Jing Liu, Xue Cui.

**Writing – review & editing:** Liang Ma, Ping Fu.

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
