## [Decision Letter · Decision Letter 0]

13 Aug 2019

PONE-D-19-18223

2-Methylquinazoline Derivative F7 as a potent and selective HDAC6 Inhibitor Protected against Rhabdomyolysis-induced Acute Kidney Injury

PLOS ONE

Dear Dr Ma,

Thank you for submitting your manuscript to PLOS ONE. After careful consideration, we feel that it has merit but does not fully meet PLOS ONE’s publication criteria as it currently stands. Therefore, we invite you to submit a revised version of the manuscript that addresses the points raised during the review process.

The major merit of the study is the synthesis of a new HDAC6 inhibitor. The author is expected to discuss the difference between their novel HDAC6 inhibitor F7 and other HDAC6 inhibitors such as ACY1215.

Please read comments from reviewers carefully and address them accordingly.

We would appreciate receiving your revised manuscript within sixty days of the date of this decision. To enhance the reproducibility of your results, we recommend that if applicable you deposit your laboratory protocols in protocols.io, where a protocol can be assigned its own identifier (DOI) such that it can be cited independently in the future. For instructions see: http://journals.plos.org/plosone/s/submission-guidelines#loc-laboratory-protocols

We look forward to receiving your revised manuscript.

Kind regards,

Tao Liu, PhD

Academic Editor

PLOS ONE

Journal Requirements:

1. To comply with PLOS ONE submissions requirements, please provide methods of sacrifice in the Methods section of your manuscript.

Reviewers' comments:

Reviewer's Responses to Questions

**Comments to the Author**

1. Is the manuscript technically sound, and do the data support the conclusions?

Reviewer #1: Yes

Reviewer #2: Yes

2. Has the statistical analysis been performed appropriately and rigorously? 

Reviewer #1: Yes

Reviewer #2: Yes

3. Have the authors made all data underlying the findings in their manuscript fully available?

Reviewer #1: Yes

Reviewer #2: Yes

4. Is the manuscript presented in an intelligible fashion and written in standard English?

Reviewer #1: Yes

Reviewer #2: Yes

5. Review Comments to the Author

Reviewer #1: In this study, Liu et al. have identified a novel HDAC6 inhibitor F7can alleviate rhabdomyolysis-induced AKI through inhibition of NF-kappB and ERK1/2 signaling pathway and reduction of IL-beta expression. This is an interesting observation and provides a novel inhibitor HDAC6. The manuscript may be improved through addressing following concerns.

1. In Figure 3D, expression of phospho-ERK1/2 and NGAL should be examined by immunoblot analysis.

2. In figure 3A, Glycerol was missed in the second picture

3. In Figure 4B, it seems that F7 treatment does not affect Mb-induced deacetylation of H3. The authors may examine whether F7 affect acetylation of alpha-tubulin in order to reflect its inhibitory effect in vitro.

4. The authors should discuss the comparison about the efficacy, chemical property or administration route etc between F7 and 23BB, another HDAC6 inhibitor synthesized by this group.

Reviewer #2: In this study, Jing Liu et al. show that a new selective inhibitor of HDAC6, synthesized by Liang Ma’s group, reduces renal injury and inflammation trigger by intramuscular glycerol injection.

I would like to ask the authors to:

1. Add in the Material and Methods the description of F7 inhibitor synthesis and of the assay performed to show its selectivity for HDAC6.

2. In figure legends, the N show be added for each figure.

3. The cytokine levels should be determined in the cell supernatant or kidney lysates by ELISA.

4. In fig 2A, can the author show the expression of other HDAC besides HDAC6, to exclude the possibility that the F7 inhibitor affect the expression of other HDACs.

5. Can the percentage of apoptotic cells in fig 4A be quantified in a graph?

6. It would be interesting to compare the efficiency of the two HDAC6 inhibitors 23BB and F7 synthesized by the group and utilized in the same AKI model. Could the authors provide some evidence (at least with in vitro assays on HK2 cells) of the superior efficacy of F7 (assumption based on IC50) in preventing tubular cell apoptosis, NFkB activation and cytokine production?

6. PLOS authors have the option to publish the peer review history of their article (what does this mean?). If published, this will include your full peer review and any attached files.

Reviewer #1: No

Reviewer #2: No

---

## [Author Response · Author response to Decision Letter 0]

26 Sep 2019

To Reviewers:

Reviewer #1: In this study, Liu et al. have identified a novel HDAC6 inhibitor F7 can alleviate rhabdomyolysis-induced AKI through inhibition of NF-kappB and ERK1/2 signaling pathway and reduction of IL-beta expression. This is an interesting observation and provides a novel inhibitor HDAC6. The manuscript may be improved through addressing following concerns.

1. In Figure 3D, expression of phospho-ERK1/2 and NGAL should be examined by immunoblot analysis.√

Thanks for your reminding. We added western blot images of NGAL and phosphor-ERK1/2 into Figure 3 (shown in Fig 3B and Fig3D respectively). Actually, we tested phospho-ERK1/2 expression using WB before and found no significant ERK1/2 phosphorylation reduction in F7 treatment group, thus only stated the changes of total ERK1/2 in three groups. Together with supplements of NGAL and p-ERK1/2, we revised the corresponding Result parts as follows:

“Renal protection by HDAC6 inhibition was further tested by real-time PCR and immunoblot analysis. NGAL mRNA expression and protein expression were both obviously elevated in glycerol group while pre-treatment of F7 markedly reduced expressions (Figure 3B).”

“Moreover, total extracellular signal-regulated kinases (ERK1/2) expression and ERK1/2 phosphorylation level was upregulated activated in the glycerol group, together with obvious p38 phosphorylation. F7 effectively inhibited total ERK1/2 expression and p38 phosphorylation (Figure 3D).”

2. In figure 3A, Glycerol was missed in the second picture.

“Glycerol” was added in Figure 3A. Thanks.

3. In Figure 4B, it seems that F7 treatment does not affect Mb-induced deacetylation of H3. The authors may examine whether F7 affect acetylation of alpha-tubulin in order to reflect its inhibitory effect in vitro.

Thanks for your suggestion. We examined acetylated alpha-tubulin expression level and found inhibitory effect from F7 was more obvious on acetylated alpha-tubulin than acetylated H3. We added the western blot image as shown in Figure 4B.

 4. The authors should discuss the comparison about the efficacy, chemical property or administration route etc between F7 and 23BB, another HDAC6 inhibitor synthesized by this group.

Thanks for your suggestions. F7 and 23BB are all 2-Methylquinazoline Derivative, which indicates that they have homologous structure, chemical property and similar HDAC inhibition effects. We supplemented a table (S1 Table) comparing F7 with other HDAC6 inhibitors using the same HDAC enzyme inhibition assay for you reference. From the table you could find F7 outperformed 23BB with better efficacy (1/3 HDAC6 inhibition IC50) and better selectivity, and the same finding could be acquired when comprehensively comparing F7 with other HDAC6 inhibitors in the table. We have already administrated 23BB on RM-AKI model before, and the purpose of this research is to verify the reno-protective efficacy and safety (shown in S2 Fig). We added the comparison between F7 and 23BB in Discussion as follows:

“HDAC enzyme inhibition assay indicated that HDAC6 inhibition efficacy of F7 was three times higher than that of 23BB, and inhibition selectivity ratio of F7 (inhibition activity of HDAC6 vs other HDACs) were all much higher than that of 23BB (details were listed in S1 Table). Base on the good performance on inhibition efficacy and selectivity, we further investigated the renoprotective effect of F7 and the involved mechanisms in rhabdomyolysis-induced AKI.”

Reviewer #2: In this study, Jing Liu et al. show that a new selective inhibitor of HDAC6, synthesized by Liang Ma’s group, reduces renal injury and inflammation trigger by intramuscular glycerol injection.

I would like to ask the authors to:

1. Add in the Material and Methods the description of F7 inhibitor synthesis and of the assay performed to show its selectivity for HDAC6.

Thanks for your suggestion. The methods of F7 synthesis and HDACs inhibition selectivity assay performance was all added in the Material and methods- Rhabdomyolysis-induced AKI model and pretreatment, shown as follows:

“The synthesis of F7 was in six synthetic steps with moderate yield according to the previous procedures of selective HDAC6 inhibitor 23bb[21]. HDACs inhibition assay of F7 was performed utilizing 4-amino-7-methylcoumarin (AMC) labeled Ac-peptide (Ac-peptide-AMC) substrates, a service provided by Chempartner Company (Shanghai, China). Results of IC50 and selectivity of F7 against HDAC1-11 enzymes were all listed in S1 Fig.”

2. In figure legends, the N show be added for each figure.

Thanks for reminding. Number of animals and repeat times were added.

3. The cytokine levels should be determined in the cell supernatant or kidney lysates by ELISA.

We tested three pro-inflammatory cytokines (TNF-α, IL-1β, and IL-6) in kidney lysates and two cytokines (TNF-α and IL-1β) in serum. Results were added in Fig 3G & Fig 3H. Corresponding Result part was revised as follows:

“Proinflammatory cytokines in kidney lysates and serum were also examined using ELISA. It could be found in both serum and kidney lysates that IL-β and TNF-α were obviously elevated in glycerol group while decreased in F7 treatment group. Additionally, similar change was observed in IL-6 from kidney lysates (Figure 3G-3F).”

4. In fig 2A, can the author show the expression of other HDAC besides HDAC6, to exclude the possibility that the F7 inhibitor affect the expression of other HDACs.

Thanks for reminding that. Before F7 treatment, we examined all the HDACs inhibition to make sure F7 selectively inhibited HDAC6 as shown in S1 Fig. F7 displayed the best selectivity of HDAC6 versus other HDACs in the form of 110-1724 times. Base on this evidence, we thought it might be not that necessary to examine other HDACs.

5. Can the percentage of apoptotic cells in fig 4A be quantified in a graph?

Apoptotic cells were all quantified and presented in the last picture of Fig 4A. Please check it.

6. It would be interesting to compare the efficiency of the two HDAC6 inhibitors 23BB and F7 synthesized by the group and utilized in the same AKI model. Could the authors provide some evidence (at least with in vitro assays on HK2 cells) of the superior efficacy of F7 (assumption based on IC50) in preventing tubular cell apoptosis, NFkB activation and cytokine production?

It’s really good suggestions, thanks. The reasons why we only used F7 are: (i) F7 and 23BB these two HDAC6 selective inhibitors are with highly structural homology, which indicates that they have similar HDAC6 inhibitory efficiency. (ii) Before F7 treatment, we compared the HDAC6 inhibitory selectivity of F7 to other common HDAC6 inhibitors (S1 Table) using the same HDAC inhibition assay method. F7 outperformed all the other inhibitor with satisfying inhibition efficacy as well as best selectivity. Therefore, the purposes of this research are to verify the reno-protective efficacy and safety, by setting Glycerol group as reference and examining other organs’ pathologic changes due to F7 utilization (S2 Fig). To be rigorous and complete, we will further administrate F7 and 23BB in the same AKI model as you suggested. Thanks. 

Comparison between F7 and 23BB were added in Discussion as follows:

“HDAC enzyme inhibition assay indicated that HDAC6 inhibition efficacy of F7 was three times higher than that of 23BB, and inhibition selectivity ratio of F7 (inhibition activity of HDAC6 vs other HDACs) were all much higher than that of 23BB (details were listed in S1 Table). Base on the good performance on inhibition efficacy and selectivity, we further investigated the renoprotective effect of F7 and the involved mechanisms in rhabdomyolysis-induced AKI.”

---

## [Decision Letter · Decision Letter 1]

8 Oct 2019

2-Methylquinazoline Derivative F7 as a potent and selective HDAC6 Inhibitor Protected against Rhabdomyolysis-induced Acute Kidney Injury

PONE-D-19-18223R1

Dear Dr. Ma,

We are pleased to inform you that your manuscript has been judged scientifically suitable for publication and will be formally accepted for publication once it complies with all outstanding technical requirements.

With kind regards,

Tao Liu, PhD

Academic Editor

PLOS ONE

Additional Editor Comments (optional):

Reviewers' comments:

Reviewer's Responses to Questions

**Comments to the Author**

1. If the authors have adequately addressed your comments raised in a previous round of review and you feel that this manuscript is now acceptable for publication, you may indicate that here to bypass the “Comments to the Author” section, enter your conflict of interest statement in the “Confidential to Editor” section, and submit your "Accept" recommendation.

Reviewer #1: All comments have been addressed

Reviewer #2: All comments have been addressed

2. Is the manuscript technically sound, and do the data support the conclusions?

Reviewer #1: Yes

Reviewer #2: Yes

3. Has the statistical analysis been performed appropriately and rigorously? 

Reviewer #1: Yes

Reviewer #2: Yes

4. Have the authors made all data underlying the findings in their manuscript fully available?

Reviewer #1: Yes

Reviewer #2: Yes

5. Is the manuscript presented in an intelligible fashion and written in standard English?

Reviewer #1: Yes

Reviewer #2: Yes

6. Review Comments to the Author

Reviewer #1: My concerns have been fully addressed. This manuscript can be accepted for publication in Plos one now

Reviewer #2: (No Response)

7. PLOS authors have the option to publish the peer review history of their article (what does this mean?). If published, this will include your full peer review and any attached files.

Reviewer #1: No

Reviewer #2: Yes: Elena Rampanelli

---

## [Editor Report · Acceptance letter]

14 Oct 2019

PONE-D-19-18223R1 

2-Methylquinazoline derivative F7 as a potent and selective HDAC6 inhibitor protected against rhabdomyolysis-induced acute kidney injury 

Dear Dr. Ma:

I am pleased to inform you that your manuscript has been deemed suitable for publication in PLOS ONE. Congratulations! Your manuscript is now with our production department. 

With kind regards,

on behalf of

Dr. Tao Liu 

Academic Editor

PLOS ONE